# Examining the Relationships between the Incidence of Infectious Diseases and Mood Disorders: An Analysis of Data from the Global Burden of Disease Studies, 1990–2019

**DOI:** 10.3390/diseases11030116

**Published:** 2023-09-06

**Authors:** Ravi Philip Rajkumar

**Affiliations:** Department of Psychiatry, Jawaharlal Institute of Postgraduate Medical Education and Research, Pondicherry 605006, India; jd0422@jipmer.ac.in; Tel.: +91-413-2296280

**Keywords:** major depressive disorder, bipolar disorder, infectious diseases, upper respiratory infections, lower respiratory infectious, lung–brain axis, gut–brain axis

## Abstract

Mood disorders are among the commonest mental disorders worldwide. Epidemiological and clinical evidence suggests that there are close links between infectious diseases and mood disorders, but the strength and direction of these association remain largely unknown. Theoretical models have attempted to explain this link based on evolutionary or immune-related factors, but these have not been empirically verified. The current study examined cross-sectional and longitudinal associations between the incidence of infectious diseases and mood disorders, while correcting for climate and economic factors, based on data from the Global Burden of Disease Studies, 1990–2019. It was found that major depressive disorder was positively associated with lower respiratory infections, while bipolar disorder was positively associated with upper respiratory infections and negatively associated with enteric and tropical infections, both cross-sectionally and over a period of 30 years. These results suggest that a complex, bidirectional relationship exists between these disorders. This relationship may be mediated through the immune system as well as through the gut–brain and lung–brain axes. Understanding the mechanisms that link these groups of disorders could lead to advances in the prevention and treatment of both.

## 1. Introduction

Major depressive disorder (MDD) and bipolar disorder (BD) are the two major types of mood disorders. Major depressive disorder, commonly referred to as depression, is the most common mental disorder globally. According to an estimate published by the World Health Organization in 2018, the point prevalence of MDD is around 4.4% of the world’s population; in other words, over 300 million people around the world suffer from depression at any given time [1]. Data from the World Mental Health Survey Initiative yielded somewhat higher estimates, with a 12-month prevalence of around 5–6% and a lifetime prevalence of 11–14% [2]. Due to its frequency, symptomatology, and tendency to recur or persist, depression is one of the leading global causes of disability [3]. A significant proportion of patients with an initial diagnosis of MDD may later be re-diagnosed as having bipolar disorder. BD is characterized by recurrent episodes of both elevated mood (mania) and depression, and it is associated with a higher level of disability than MDD [4]. Though BD and MDD are considered to be distinct disorders in current classificatory systems, there is a substantial clinical, genetic, and neurobiological overlap between them, and about 22% of patients with MDD are later diagnosed as having BD [5,6,7,8,9]. There are significant cross-national variations in the epidemiology of both MDD and BD, with higher prevalence estimates being reported in high-income countries than in low- and middle-income countries [2,10]. Several biological, psychological, and social explanations for this variation have been proposed, but none of them has been conclusively established [11,12,13,14,15,16,17,18,19,20].

Over the last two decades, increasing attention has been paid to the relationship between infectious diseases and mood disorders, with a specific focus on certain types or routes of infection. Though mood disorders are not infectious diseases per se, the existing literature suggests that there are significant evolutionary, mechanistic, and epidemiological links between these disease groups.

### 1.1. Evolutionary Links between Infectious Diseases and Mood Disorders

Culture-gene co-evolution: One of the first attempts to link infectious diseases to mood disorders from an evolutionary standpoint was performed by Chiao and Blizinsky [21]. According to their hypothesis, evolutionary pressures caused by infectious pathogens led to a phenomenon of “culture-gene co-evolution”. This phenomenon has two components: (a) selection for specific alleles, such as the *s* allele of the serotonin transporter functional polymorphism *5-HTTLPR*, and (b) the development of collectivist cultural values that would reduce infectious disease morbidity through the enforcement of strict in-group norms and reduced contact with “outsiders”. The benefits conferred by this process were two-fold: improved survival in areas with a high pathogen load and improved mental health. In regions where there was a mismatch between genotype and culture (e.g., where the *5-HTTLPR s* allele existed along with individualistic cultural values), depression would result. In support of this hypothesis, the authors noted an inverse correlation between *s* allele distribution and the national prevalence of depression, and a similar inverse relationship between cultural collectivism and the prevalence of this disorder. This finding was subsequently replicated by other authors and extended to other genetic variants, such as functional polymorphisms of the *MAOA* and *OPRM1* genes [22]. In this model, the link between infectious diseases and mood disorders is indirect and remote, and no association between the current distributions of these groups of disorders is inferred.Depression as an evolved defense mechanism against infection: Subsequently, researchers have considered the possibility of a more direct evolutionary relationship between mood disorders and infectious diseases. This is based on observed similarities between the symptoms of depression and “sickness behavior”, which has evolved as a defense against infectious diseases. Sickness behavior, which is mediated by cytokines such as interleukin-6 (IL-6) and tumor necrosis factor alpha (TNF-α) includes reduced appetite, fatigue, reduced physical activity, and social withdrawal. This set of responses is adaptive both at the individual level (by facilitating tissue healing and recovery from infectious diseases) and at the group level (by minimizing transmission to other members of the same species) [23,24,25,26]. According to this model, depression persists in human populations, despite its obvious disadvantages, because it is fundamentally similar to sickness behavior. If this model were correct, one would expect to find an inverse relationship between depression and infectious diseases both in individuals and in communities. However, this has not been observed in real-world settings, in which depression is associated with increased rates of certain infectious diseases [27].The hygiene hypothesis: According to the “hygiene hypothesis” or “Old Friends hypothesis”, exposure to certain infectious agents, such as helminths and saprophytic mycobacteria, is important for optimal regulation of the immune system. In other words, a mutually beneficial relationship has evolved between these infectious organisms and the human species. In modern societies characterized by high levels of hygiene and sanitation, exposure to these organisms is reduced or absent, leading to a loss of immune regulation and tolerance, leading to increased basal cytokine levels and exaggerated immune responses to antigens. This is associated with an increased risk of allergic diseases (such as asthma), immune-inflammatory disorders (such as inflammatory bowel disease), and mood disorders [28,29]. If this hypothesis was correct, one would expect to find lower levels of mood disorders in low- and middle-income countries with higher rates of intestinal infections and a positive correlation between levels of hygiene and depression. The former point has been observed to some extent in global surveys, such as the World Mental Health Survey and Global Burden of Disease studies [1,2,30]. However, it is contradicted by the observation that in developing countries, poor sanitation is positively correlated with levels of depression [31]. In light of such findings, proponents of this hypothesis have suggested that the association between hygiene and depression is specific to high-income countries and may be related to the concurrent effects of urbanization, socioeconomic inequality, and a lack of exposure to “green spaces”. However, these proposals have not yet been tested [32].Depression as a heterogeneous response to stress: The most recent evolutionary model of mood disorders, proposed by Rantala et al. [33], posits that depression is not a single entity, but a group of related conditions arising in response to specific triggers. This model enumerates 12 triggers that could lead to depression, one of which is infection. According to these researchers, depression triggered by infection is an adaptive response whose function is to reduce the risk of disease transmission to others, and to conserve energy for the proper functioning of the immune system. On the other hand, the triggering of immune-inflammatory responses by non-infectious factors, such as stress and dietary factors, may lead to bipolar disorder [34]. If this model were correct, one would expect to find a positive association between infectious diseases and depression, but not for bipolar disorder.

### 1.2. Mechanistic Models Linking Infectious Diseases and Mood Disorders

The true nature of the evolutionary links between infection and mood disorders, if any, remains an open question. However, even if these hypotheses are set aside, there are certain mechanistic pathways that overlap between mood disorders and various kinds of infectious diseases. It is well documented that mood disorders are associated with elevated levels of certain immune-inflammatory mediators, even in the absence of an identifiable infection [35,36]. This finding has led various researchers to speculate on possible links between infection, immunity, and symptoms of mood disorders.

PATHOS-D: The PATHOS-D (Pathogen Host Defense) hypothesis, developed by Raison and Miller, is probably the best-characterized mechanistic model linking infection and depression [37,38]. This model highlights the fact that genetic risk alleles for depression are associated with immune responses to infection, including those involved in the induction of sickness behavior. In this model, the role of exposure or non-exposure to specific pathogens is not central. Instead, emphasis is placed on the fact that genetic variants that were adaptive in defenses against infectious disease (in “ancestral” environments) can be associated with increased immune-inflammatory activity in response to non-infectious social stressors, leading to the maladaptive phenotype of MDD in “modern” environments. This hypothesis is more biologically plausible than the earlier “Old Friends” model [27] and has the advantage of being consistent with existing evidence on the relationships between genotype, stressors, and vulnerability toward MDD [39,40,41,42,43]. If this model was true, one might expect to find significant correlations between the prevalence of infectious diseases and mood disorders, but not a causal relationship. Some authors have attempted to link the PATHOS-D model and the hygiene hypotheses by postulating a two-stage process. In the first step, exposure to “tolerogenic” microorganisms, or the lack of such exposure, influences host immune responses to stress; in the second, psychosocial adversity triggers mood disorders—and more specifically depression—at higher rates in those without such a beneficial exposure [32,44,45]. This proposal, though interesting, has not been formally evaluated in cross-regional or cross-national research. If it were true, one would expect to find shifts in the incidence of mood disorders with increasing levels of “hygiene” after correcting for the effects of exposure to stress.Gut–brain axis models: Research over the past three decades has found evidence of extensive “crosstalk” between the gut and the central nervous system, mediated through a variety of signaling pathways, namely the parasympathetic nervous system, the gut microbiota, gut hormones, and the immune–inflammatory–stress axis [46,47,48]. The gut–brain axis can link infectious diseases and mood disorders through at least three mechanisms. First, intestinal infections can directly lead to symptoms of depression by altering neurotransmitter, hormonal, or inflammatory signaling between the gut and brain [49]. In this case, one would expect to find a positive association between specific intestinal infections and mood disorders. Second, changes in gut microbiota—caused either by infection, diet, or changes in the level of hygiene—can alter stress sensitivity, thus affecting an individual’s vulnerability to mood disorders [50]. If this possibility were true, one would expect findings similar to those of the hygiene hypothesis. Third, treatment of infectious diseases with antibiotics can alter the composition of the gut microbiome. This leads to downstream effects on the stress response and emotional processing that can predispose to mood disorders [51,52]. In this case, the primary trigger would be antibiotic use and not infection, and a positive correlation between antibiotic usage patterns and mood disorders would be expected.

On reviewing the evolutionary and mechanistic models linking infectious and mood disorders, a certain overlap between them is evident. All of them assign a central role to host immune-inflammatory responses and how these responses are influenced either positively or negatively by exposure to specific micro-organisms. This is depicted graphically in Figure 1, as shown below. These models are biologically plausible but stand in need of verification through empirical research. Moreover, most of these models focus largely on MDD and not on BD, despite consistent evidence of a close link between these disorders. From an evolutionary perspective, some aspects of BD may have emerged as a natural defense or “override mechanism” to protect individuals and groups from the negative effects of prolonged depression [53]. Building on this perspective, hypomanic “defenses” may have been particularly effective in those ancestral environments where the burden of infectious disease was relatively low. In these environments, prolonged “sickness behavior”-like phenotypes may have been maladaptive and interfered with other behaviors essential for survival, such as foraging for food, social interactions, and reproduction [54]. Thus, any evolutionary or mechanistic model that attempts to link infectious diseases and MDD should also take BD into consideration. Though none of these models have been formally tested on a large scale, they have some support from epidemiological and clinical research linking infection and mood disorders.

### 1.3. Epidemiological and Clinical Links between Infectious Diseases and Mood Disorders

General considerations: Certain studies support a non-specific association between infectious diseases and the subsequent development of mood disorders. A meta-analysis of 28 studies examining the association between infectious diseases and mood disorders found positive associations between several infectious organisms and MDD [55]. A study of youth in the United States found that infectious diseases in childhood were associated with an almost four-fold increase in the risk of MDD [54]. In a similar study from Taiwan, exposure to bacterial infections in childhood was associated with a 2-fold increase in MDD and a 2.5-fold increase in BD. The 11 causative organisms evaluated in this study included common pathogens such as *Staphylococcus*, *Streptococcus*, *Hemophilus,* and *Pseudomonas* [56]. Two Danish studies also found a significant association between overall infection load and mood disorders [57,58]. These results, though requiring replication in low- and middle-income country settings, suggest that infectious diseases may, at the very least, act as a trigger for mood disorders in vulnerable individuals [59].Respiratory infections: In the meta-analytic review cited above, significant associations with MDD were reported chiefly for airborne viruses [55]. Anecdotal evidence linking respiratory infections and depression antedates the formulation of specific evolutionary or immunological hypotheses by several decades. For example, episodes of depression were observed following influenza and appeared frequently in works of fiction written following outbreaks of this disease [60,61]. These literary observations were supplemented by clinical case reports published in the 1970s and 1980s, linking influenza to the subsequent onset for both MDD and BD [62,63,64,65,66,67]. Subsequently, an analysis of data from over 100,000 patients confirmed the association between influenza and MDD; recent influenza was associated with a 1.5-fold increase in subsequent MDD, and recurrent infections appeared to increase this risk further [68]. Similar findings have been reported in patients following recovery from other viral respiratory infections, including SARS [69] and COVID-19 [70,71]. The presence of antibodies against influenza viruses or coronaviruses has also been non-specifically associated with an increase in the risk of mood disorders [72]. Studies from southern African countries have also reported associations between pulmonary tuberculosis and subsequent MDD [73,74]. Thus, both upper and lower respiratory infections appear to be associated with mood disorders.Intestinal infections: Due to their potential effects on mood through alterations in the gut–brain axis, enteric infections have also been evaluated as potential triggers or risk factors for MDD and BD. A study comparing patients with MDD and healthy controls found higher levels of IgM and IgG antibodies against gram-negative enterobacteria in the depression group [75]. A prospective study of patients with intestinal infections (bacterial, viral, or parasitic) found that this group of diseases was associated with a two-fold increase in both BD and MDD. The mean time lag between infection and the onset of mood disorder in this sample was 2.4 years, and this association remained significant after correction for sociodemographic variables [76]. Several case reports have suggested a possible causal association between enteric (typhoid) fever and both BD and MDD [77,78,79]. A recent study from Egypt extended these findings by documenting MDD in asymptomatic carriers of *Salmonella typhi*, which improved after antibiotic therapy without a need for concurrent antidepressant medication [80]. In addition, a study of psychiatric inpatients from Ethiopia found evidence of intestinal parasitic infections in 15% of patients with MDD and 9% of patients with BD [81].Tropical infections: Symptoms of MDD and BD in patients with tropical infections have been documented for over a century. Such symptoms have often been viewed as psychogenic in origin, but direct biological effects are an equally plausible explanation [82,83]. A recent meta-analytic review found that certain tropical parasitic diseases were associated with mood disorders. Chagas disease and cysticercosis were specifically linked to high rates of depressive symptoms, while toxocariasis was associated with BD [84]. Though these results could be interpreted as reflecting the neurotropic tendencies of these parasites, peripheral mechanisms cannot be ruled out [85]. Malaria, the most common tropical infection worldwide, has also been associated with MDD even in cases where there is no evidence of brain involvement [86,87]. High levels of depressive symptoms have also been observed in patients with lymphatic filariasis from several African countries [88,89,90]. Though these symptoms have been ascribed to the effects of deformities or disability caused by lymphedema, they may also reflect the immune-inflammatory changes associated with this disease [91,92]. Experts in the field have also highlighted the close associations between tropical infections and mental disorders, such as MDD, and the need to elucidate the mechanisms linking them [93].Vaccine-preventable diseases: Common vaccine-preventable diseases, such as measles, pertussis, poliomyelitis, and rubella, have seen a resurgence in the past decade in both developed and developing countries [94]. There is some evidence that infections in this group may be linked to both MDD and BD. Elevated levels of measles antibodies have been identified in patients with recent-onset, severe (“psychotic”) episodes of both MDD and BD [95], and measles antibody titers were correlated with a measure of intestinal inflammation in patients with BD [96]. A more recent study also found a marginal association between BD and higher measles IgG antibody titers [97]. Though no such direct research has been conducted for other vaccine-preventable diseases, there is indirect evidence linking them to mood disorders. Specifically, immunization with a live attenuated rubella vaccine has been associated with symptoms of MDD in girls from a low socioeconomic background [98], and increased pertussis toxin-activated G protein ribosylation has been documented in postmortem brain tissue from patients with BD [99].

### 1.4. Summary and Rationale for the Current Study

Both theoretical and empirical considerations support the notion of a close link between various groups of infectious diseases and mood disorders. Given the complex and competing nature of the various proposed hypotheses and the preliminary nature of most clinical and epidemiological findings, it would be useful to examine available data on the relationships between specific infectious diseases and the epidemiology of mood disorders. The Global Burden of Disease (GBD) studies, conducted over the period 1990–2019, provide estimates of the incidence, prevalence, and disability associated with a wide range of diseases across all the countries and regions of the world [100]. Examination of GBD data has revealed a reduction in the burden of infectious diseases over the past two decades, with a contemporaneous increase in the burden associated with depression [3,101]. This phenomenon has been referred to as an “epidemiological transition” but has not specifically been evaluated in the context of hypotheses such as PATHOS-D or the hygiene hypothesis [102]. Examining both cross-sectional and longitudinal associations between these groups of disorders could help refute, confirm, or refine existing theoretical models of the links between infectious diseases and mood disorders or even yield new results that could deepen our understanding of these links.

The aim of the current study was to examine associations between the incidence of infectious diseases, both as a group and in specific categories, and the incidence of MDD and BD over the past 30 years, based on GBD data. This study was exploratory in nature and was not based on a specific theoretical model. However, its results will be examined in terms of how well they align with each model.

## 2. Materials and Methods

The current study was a country-level association study employing both cross-sectional and longitudinal analyses. The estimates of disease incidence provided by the Global Burden of Disease studies, 1990–2019, constituted the primary data source for these analyses. The methodology of the GBD studies has been described in detail by the researchers and involves collection of data from a wide range of sources, including government and hospital records, registries for specific diseases, and published data in journals, books, or online databases. This information is subjected to a rigorous process of cleaning, aggregation, and modeling to yield estimates of disease incidence, prevalence, and disability-adjusted life years (DALYs) for each country and region [103]. GBD data are freely available for use for academic or research purposes and can be obtained through database queries from the website of the Institute for Health Metrics and Evaluation (IHME), which is the primary coordinating agency for the GBD studies [104].

### 2.1. Data Sources

Information on the incidence of infectious diseases and MDD for the period 1990–2019 was obtained from the GBD database [105]. Incidence was selected as the measure of interest to minimize any possible confounding effect arising from chronic or recurrent cases of MDD or BD [106,107]. To reduce the possibility of a confounding effect of life expectancy or age distributions across countries [108], only age-standardized estimates of incidence were retrieved for each category.

For infectious diseases, age-standardized estimates were obtained for the following categories: “upper respiratory infections”, “lower respiratory infections”, “enteric infections”, “neglected tropical diseases and malaria”, and “other infectious diseases”. These categories, which were provided by the GBD database, were selected based on existing theoretical and empirical evidence, as detailed in the Introduction. The group “other infectious diseases” in the GBD dataset corresponded approximately to the group of “vaccine-preventable diseases” such as measles, pertussis, and rubella. Data on MDD, BD, and infectious diseases were available for 204 countries.

### 2.2. Confounding or Interacting Factors

Cross-national variations in the epidemiology of MDD and BD may be due to a variety of factors, which may operate either independent of infectious-immune-inflammatory mechanisms or synergize with them. These factors include climate, urbanization, and economic factors such as overall economic development and income inequality [109,110,111]. Some of these factors may also influence the epidemiology of infectious diseases; for example, tropical and related diseases are more common in countries located close to the equator [112], while higher levels of human development may be associated with lower levels of intestinal infections [113]. To address the possibility of such interactions or confounding effects, data on the following variables was also collected for the years 1990 and 2019:The Human Development Index (HDI), a composite measure of life expectancy, education, and per capita income, as obtained from official United Nations publications [114,115]. This information was available for 124 countries in 1990 and for 187 countries in 2019.Levels of urbanization in each country, as measured by the percentage of the total population residing in urban areas, provided by the World Bank’s database [116]. This information was available for 200 countries at both time points.The Gini coefficient, a measure of income inequality at a national level, from the World Bank’s database [117]. This information was available for 168 countries at both time points.The average distance of each country from the equator, a proxy measure of climate that has been used in prior epidemiological studies of mental disorders [118,119]. This information was available for all the 204 countries included in this study.

### 2.3. Data Analysis

Data on the incidence of each group of disorders were obtained for the years 1990, 1995, 2000, 2005, 2010, 2015, and 2019. Data were retrieved at 5-year intervals in order to achieve an adequate balance between type I and type II errors in the analyses. Examination of annual rate-of-change data from the GBD database revealed that these rates were low; therefore, analyzing annual data would increase the amount of “noise” in the data set, and conducting repeated correlation analyses would increase the likelihood of false-positive findings. On the other hand, though the 30-year interval between 1990 and 2019 is appropriate for cross-lagged panel analyses [120], it is insufficient for other longitudinal analysis methods. Given that the GBD data are compiled at approximately 5-year intervals, this sampling interval was selected for the current study.

All study variables were tested for normality using the Shapiro–Wilk test. As none of the study variables conformed to a normal distribution, non-parametric statistical tests were used for all further analyses.

In the cross-sectional analyses, Spearman’s correlation coefficient was computed to examine the strength and direction of possible monotonic associations between the incidence of each type of infectious disease and the incidence of MDD and BD. To test for the possible confounding or interacting effects of socioeconomic variables and climates, correlations between each of these variables and the prevalence of infectious diseases, MDD, and BD were also calculated. Bonferroni’s correction for multiple comparisons was applied for all these analyses. If any socioeconomic or climatic variable was significantly associated with either infectious diseases or mood disorders, Spearman’s partial correlations were computed. This was performed to assess if any associations between infectious diseases and mood disorders remained significant when correcting for these variables.

For the longitudinal analyses, four methods were used. First, cross-lagged correlation coefficients were computed to test for the possibility of a causal relationship between changes in the incidence of infectious diseases and changes in the incidence of mood disorders [120]. Second, the percentage of change of each disorder for the period 1990–2019 was computed, and correlations between these measures of change were computed as for the cross-sectional analyses. To rule out any possible confounding or interacting effects, these tests were also carried out for socioeconomic variables and climate. Third, the chi-square test was used to compare the directions of change in the incidence of mood disorders based on changes in the incidence of each type of infectious disease. Finally, a general linear model was used to examine if changes in the incidence of infectious diseases were associated with trends in the incidence of mood disorders. Given that the current data violated the assumption of sphericity, a repeated-measures analysis of variance with a Greenhouse-Geisser correction was used for these analyses.

Studies of cross-regional or cross-national correlations have sometimes reported nonlinear patterns of association [121]. In order to assess this possibility, the curve estimation function of the Statistical Package for Social Sciences (SPSS, version 27.0) was used to test for the possibility of logarithmic, quadratic, or cubic associations between the incidence of infectious diseases and mood disorders.

All statistical tests were two-tailed, and a *p* value of <0.05 was used as the threshold for statistical significance. When assessing the strengths of bivariate and partial correlations, standard guideline values for biomedical research were used as follows: ρ ≥ 0.8, strong correlation; ρ = 0.6–0.79, moderate correlation; ρ = 0.3–0.59, fair correlation; ρ = 0.1–0.29, weak correlation [122].

The current study included data on infectious diseases and mood disorders for 204 countries. In order to identify at least a fair degree of correlation (regression coefficient = 0.3 or more) between these groups of diseases, with a power of 90% and a significance level of *p* < 0.05, the minimum calculated sample size was 46. Thus, this study was adequately powered to identify significant associations between both disease groups.

## 3. Results

### 3.1. Incidence of Infectious Diseases and Mood Disorders in 204 Countries, 1990–2019

Data on the incidence of infectious diseases and mood disorders were available for a total of 204 countries. These data, as well as the available data on confounding factors, are summarized in Table 1. The incidence of MDD, reported as the median and inter-quartile range (IQR) per 1000 population, was 7.48 (3.05) in 1990 and 7.10 (3.09) in 2019. The incidence of BD was 0.10 (0.04) at both time points.

When examining changes in the incidence of each disorder with time, it was noted that the incidence of MDD decreased significantly over the period 1990–2019 (Wilcoxon’s W = 15,547.5; *p* < 0.001), with a rank-biserial correlation of 0.50, indicating a medium effect size. The incidence of BD did not change significantly in either direction during this period (W = 9039.5, *p* = 0.464). For infectious diseases, there were significant decreases in the incidence of all categories of disease, except for enteric infections, which showed a significantly large increase over this period (W = 3686.5; *p* < 0.001; rank-biserial correlation = −0.65).

### 3.2. Cross-Sectional Associations between the Incidence of Infectious Diseases and Mood Disorders

Cross-sectional correlations between the incidence of each category of infectious diseases and the incidence of both types of mood disorder were calculated at 5-year intervals for the period 1990–2019. These correlations are presented in Table 2. There was no significant correlation between the incidences of MDD and of any group of infectious diseases at any time point. On the other hand, the incidence of BD was positively correlated with the incidence of URI and negatively correlated with LRI, enteric, tropical, and other infectious diseases. These correlations were significant after Bonferroni’s correction for a 7 × 7 table and ranged from weak to fair in magnitude (absolute value of ρ = 0.29 to 0.51). Scatter plots of these correlations are provided in Appendix A.

Correlations involving possible confounding or interacting variables are presented in Appendix A. For the year 1990, there was a weak positive correlation between the incidence of MDD and distance from the equator, while the incidence of BD was positively correlated with urbanization and the HDI. The same pattern of associations was observed for 2019, except that the incidence of BD in 2019 was positively correlated with distance from the equator. Only the associations between BD, urbanization, and the HDI remained significant after using the Bonferroni correction. Among infectious diseases, the incidence of upper respiratory infections was positively correlated with HDI and urbanization and negatively correlated with distance from the equator and the Gini coefficient. The reverse pattern (negative correlations with HDI and urbanization, and positive correlations with the Gini coefficient and distance from the equator) was observed for all other groups of infectious diseases.

As these findings suggested possible confounding effects, partial correlations were computed for all the associations presented in Table 2, correcting for all four of these variables. These results are presented in Table 3. It can be noted that after adjusting for economic variables and climate, the incidence of MDD was weakly and negatively correlated with the incidence of enteric infections, tropical diseases, and other infectious diseases in 1990, but not in 2019. The association between BD and upper respiratory infections was no longer significant after these adjustments, but the incidence of BD remained negatively correlated with enteric infections, tropical diseases, and other infectious diseases both in 1990 and in 2019.

### 3.3. Longitudinal Analyses

#### 3.3.1. Cross-Lagged Regression Analyses

To test for the possibility of a causal relationship between changes in the incidence of infectious diseases and changes in the incidence of mood disorders, cross-lagged regression analyses were carried out for each category of infectious disease. The initial analysis included all 204 countries. However, the accuracy of this analysis was negatively affected by the fact that 68 of these countries had values of zero or close to zero for the incidence of some infectious diseases. To address this limitation, a second analysis was carried out with these countries excluded. These results are presented in Table 4 In these analyses, no evidence was found for a significant causal relationship—in either direction—between any category of infectious disease and mood disorders. A weak trend (*p* < 0.2) was found for a possible causal association between LRI and MDD in both analyses.

#### 3.3.2. Relationships between Changes in the Incidence of Infectious Diseases and Mood Disorders over Time

Over the period 1990–2019, the incidence of MDD decreased by a median of −3.72% (IQR 10.56; range −35.16% to 34.32%), while the median percentage change for the incidence of BD was zero (IQR 6.26; range −11.54% to 17.24%). Correlations between the percentage changes in the incidence of MDD and BD, as well as the corresponding changes in the incidence of infectious diseases for the period 1990–2019, are presented in Table 5. Changes in the incidence of MDD were positively correlated with changes in both upper and lower respiratory infections and negatively correlated with changes in enteric and tropical infections. However, all of these correlations were weak (ρ < 0.3) and did not remain significant after correction for multiple comparisons, except in the case of lower respiratory infection. In contrast, changes in the incidence of BD were strongly positively correlated with changes in upper respiratory infections (ρ = 0.92, *p* < 0.001) and strongly negatively correlated with changes in enteric infections (ρ = −0.82, *p* < 0.001). Both of these associations remained significant when correcting for multiple comparisons.

When these correlations were adjusted for changes in urbanization and economic inequality over time, the basic pattern of associations remained unchanged, and changes in the incidence of BD remained strongly correlated with changes in upper respiratory (ρ = 0.88, *p* < 0.01) and enteric (ρ = −0.78, *p* < 0.01) infections.

#### 3.3.3. Categorical Associations between Changes in the Incidence of Infectious Diseases and Mood Disorders over Time

Comparisons of changes in the incidence of each mood disorder against changes in infectious diseases were also performed using these variables. The complete details of these analyses are presented in Table 6. In these analyses, increases in the incidence of MDD were more common in countries with a corresponding increase in upper respiratory infections over the study period (36/84 vs. 29/120; χ^2^ = 7.95, *p* = 0.005). No other significant association was observed for MDD. Increases in the incidence of BD were more common in countries with an increase in upper respiratory infections (78/84 vs. 18/120; χ^2^ = 120.22, *p* < 0.001), a decrease in enteric infections (50/55 vs. 46/149; χ^2^ = 58.12, *p* < 0.001), or a decrease in tropical infections (82/157 vs. 14/47; χ^2^ = 7.31, *p* = 0.007).

#### 3.3.4. General Linear Model Analyses

To test for the possibility of a time-by-infectious disease interaction for changes in the incidence of MDD and BD, general linear models using a repeated measures analysis of variance, with changes in infectious disease as the between-subject factor, were computed with appropriate corrections for deviations from the sphericity assumption. The results of these analyses are presented in Table 7. A significant time effect was observed for MDD, with the incidence of this disorder decreasing over the period 1990–2019, replicating the findings reported in Table 2. However, there were no interaction effects for changes in the incidence of any group of infectious diseases. For BD, there were significant interaction effects for upper respiratory infections, lower respiratory infections, enteric infections, and other infectious diseases. This finding implies that changes in the incidence of these groups of infectious significantly influenced changes in the incidence of BD over the period being studied. For upper respiratory and enteric infections, these interactions were greater than the main effect of time. Post-hoc testing revealed that increases in upper respiratory infections were associated with increases in BD over time, while decreases in enteric, tropical, and other infections were associated with increases in BD (*p* < 0.001 for all groups, Tukey’s test).

### 3.4. Subgroup Analyses

#### 3.4.1. Analyses of Tropical Infections

An examination of the incidence data used in this study shows that several countries had estimated incidence rates of zero or close to zero for tropical infections. As the presence of these values could have affected the validity and precision of the pertinent bivariate analyses, a sub-group analysis was carried out for tropical infections after removing all cases with a negligible estimated incidence.

After the removal of zero or near-zero values, data on 144 countries were analyzed. The incidence of tropical infections was not significantly correlated with the incidence of MDD but was negatively correlated with the incidence of BD (1990: ρ = −0.36, *p* < 0.001; 2019: ρ = −0.45, *p* < 0.001). After adjusting for confounders, the association with BD remained significant only in 2019 (ρ = −0.35, *p* < 0.001). Cross-lagged panel analyses did not reveal possible causal associations for any disorder. Correlations of percentage changes showed that changes in tropical infections from 1990 to 2019 were negatively correlated with changes in the incidence of both MDD (ρ = −0.23, *p* = 0.005) and BD (ρ = −0.43, *p* < 0.001). However, after adjusting for economic variables, only the association with BD remained significant (ρ = −0.36, *p* < 0.001).

#### 3.4.2. Analyses of Possible Interactions between Groups of Infectious Diseases

When analyzing the data on the incidence of specific subgroups of infectious diseases, it was found that there were several significant correlations between them, as can be noted in Appendix A. It can be observed that the incidence of upper respiratory infections was moderately to strongly negatively correlated with the incidence of all the other infectious disease subgroups (ρ = −0.44 to −0.91, *p* < 0.001), while the incidence of other groups of infectious diseases showed positive correlations with each other.

In the light of this finding, the analyses reported in Section 3.2 and Section 3.3.2 were repeated while controlling for the incidence (or change in incidence) of other groups of infectious diseases. After corrections for multiple comparisons, only two associations remained significant in these secondary analyses: a positive correlation between changes in upper respiratory infections and BD, and a positive correlation between changes in lower respiratory infections and MDD. A complete set of these analyses can be found in Appendix A.

### 3.5. Non-Linear Curve Fitting

To test for the possibility of non-linear associations between mood disorders and infectious diseases, non-linear curve fitting was attempted. In view of the very similar linear correlations at each time point observed in Table 3, curve fitting was carried out only for the years 1990 and 2019. If a particular model was identified consistently at both time points, it was taken to be the best possible fit; if none of them fit better than a linear regression, the latter was retained as the best fit. The results of these analyses are presented in Table 8. For MDD, the linear association was the best fit except for lower respiratory infections, where a better but non-significant fit was obtained for a cubic curve. For BD, quadratic associations were observed for upper respiratory and enteric infections, and logarithmic associations were observed for lower respiratory and tropical infections. However, the improvement in precision with these models over linear or monotonic models was marginal, and a linear model was retained for other infectious diseases.

## 4. Discussion

The current study examined cross-sectional and longitudinal associations between the incidence of specific groups of infectious diseases and the incidence of mood disorders, including both major depressive disorder (MDD) and bipolar disorder (BD). The categories of infectious diseases analyzed included upper respiratory infections (URI), lower respiratory infections (LRI), enteric infections, tropical infectious diseases, and other infectious diseases. The “other” category consisted of vaccine-preventable diseases such as diphtheria, pertussis, tetanus, and rubella (measles) [65]. In the time period covered by this study (1990–2019), spanning 30 years, there was a slight but significant decrease in the incidence of MDD but essentially no change in the incidence of BD. All categories of infectious disease showed a significant decrease in incidence, except for enteric infections, where a slight increase was observed (Table 1).

In the cross-sectional analyses, there were no significant correlations between the incidence of MDD and any infectious disease category. However, BD showed a specific pattern characterized by positive correlations with the incidence of URI and negative correlations with the other five infectious disease categories (Table 2). This pattern was stable over time and robust to correction for multiple comparisons. When correcting for possible confounding factors, no consistent pattern of associations was observed for MDD, but BD remained negatively correlated with the incidence tropical and other infectious diseases over time (Table 3). When correcting for inter-correlations between categories of infectious disease, a negative correlation between BD and enteric infections was observed at both time points; associations for MDD were weak or inconsistent.

When examining longitudinal associations, four distinct methods were used: cross-lagged correlation analyses, correlations of percentages changes, categorical associations, and general linear models. The first of these methods did not yield any evidence of a causal association between infectious diseases and either MDD or BD. The second found evidence of (a) a positive association between changes in LRI and changes in MDD, (b) a positive association between changes in URI and in BD, and (c) a negative association between changes in enteric infections and BD, even after correcting for changes in socioeconomic factors, changes in other infectious disease categories, and multiple comparisons. The third found (a) a positive association between changes in the incidence of URI and changes in the incidence of both MDD and BD and (b) negative associations between changes in the incidence of BD and changes in both enteric and tropical infections. The fourth found evidence of significant time-by-infectious disease interactions between BD and all classes of infectious disease; these were positive in the case of URI and negative for all other types of infections.

Attempts were also made to test for non-linear associations between mood disorders and infectious diseases. In the case of MDD, no significant associations were identified using the logarithmic, quadratic, or cubic curves. For BD, there was evidence that logarithmic or quadratic curve estimations were slightly superior to linear models for respiratory, enteric, and tropical infections.

These results are discussed under four main headings: comparisons between the current study’s results and those of other published research; the differential associations of infectious diseases with each type of mood disorder; the relationship of these results to existing hypotheses such as the hygiene hypothesis or PATHOS-D; and possible mechanisms underlying these associations. Following this, the strengths and limitations of the current study and the broader implications of its results are briefly outlined.

### 4.1. Comparisons with the Existing Literature

The current study suggests that statistically significant relationships exist between the incidence of specific groups of infectious diseases and the incidence of mood disorders. Though this study is the first to use global epidemiological data to investigate this relationship, several other researchers have examined the links between infection and mood disorders using other methods. A study of children and adolescents in Taiwan found that bacterial infections were associated with a 2.5-fold increase in the risk of BD and a 2-fold increase in the risk of MDD, with the strongest associations being reported for streptococcal infections [57]. However, this study’s findings were not specific to mood disorders: elevated rates of subsequent diagnoses of autism spectrum disorder, attention-deficit/hyperactivity disorder, and obsessive-compulsive disorder were also associated with bacterial infection in this sample. A larger study from the same country, including both children and adults and focusing exclusively on intestinal infections, reported similar results; intestinal infections were associated non-specifically with a wide range of mental disorders, including a 1.5–2-fold increase in mood disorder risk [76]. This non-specific pattern of association was also observed in a study from the United States, where childhood infections were not only associated with an increased risk of MDD but also of anxiety disorders and oppositional-defiant disorder (ODD) [56].

These relationships are not unidirectional. For example, a third Taiwanese study found a prospective association between a diagnosis of BD and the subsequent risk of both URI and LRI. This association was not specific to infectious diseases and involved cardiovascular, cerebrovascular, and non-infective gastrointestinal disorders [123]. A study of hospitalizations among patients with BD and related disorders in the United Kingdom found that this disorder was associated with an increased risk of hospitalization for infectious diseases in general, as well as for non-infectious respiratory, digestive, and urinary disorders [124]. A follow-up study of cohort of German patients found that MDD was associated with increased rates of LRI and intestinal infections, with a “dose-response” relationship between the severity of MDD symptoms and the risk of these infections [125]. Though the current study did not test these causal relationships in the other direction, it is likely that mood disorders may both be triggered by infections and increase vulnerability to subsequent infections.

More recent studies in larger cohorts or biobank samples, using genetic markers or risk scores, have underlined the complexity of the association between infectious diseases and mood disorders. In a Mendelian randomization study of over 300,000 participants in the UK Biobank, the genetic risk score for MDD was significantly associated with enteric infections caused by *Escherichia coli* [126]. Complementary to this result, two studies in a Danish cohort of over 65,000 participants, involving both epidemiological and genetic data, found a positive association between infection load and mood disorders. These studies also suggested that this association might be genetically mediated in the case of MDD, but not of BD [58,59].

In addition to this research on the broad categories of infectious disease, some researchers have examined the links between infections with specific pathogens and mood disorders. For example, a prospective relationship has been documented between Epstein-Barr virus (EBV) infection, causing infectious mononucleosis, and MDD both in the short and long term [127]. Similar associations with MDD have been reported for influenza virus infections [68] and pulmonary tuberculosis [128].

The results of the current study are partly consistent with these results. In both cases, positive associations between certain types of infectious diseases and mood disorders, particularly respiratory infections, have been documented. However, the exact direction of causality cannot be inferred from our data. The one divergent finding was that of a positive association between enteric infections and both MDD and BD in a Taiwanese sample; in the current study, enteric infections were negatively associated with BD. This may reflect differences in levels of analysis; an association that is negative at the cross-national level may be positive in a specific country or region due to the confounding effects of both population genetics and environmental factors, such as diet or social support [10,22].

### 4.2. Differential Associations of Infectious Diseases with Major Depression and Bipolar Disorder

This study’s findings suggest that there is a differential association between infectious disease groups and distinct categories of mood disorders. In cross-sectional analyses, significant and consistent correlations were observed only for BD. In longitudinal analyses, disorder-specific patterns of association emerged: MDD was positively associated with LRI, while BD was positively associated with URI and negatively associated with enteric and tropical infections.

These findings differ from those of the cohort studies discussed above, which suggest that MDD is associated with enteric infections and is more strongly linked to infectious diseases than BD. Differences in the levels or units of analysis, as mentioned in the previous section, may account for some of these differences, but it is also possible that specific pathogens may interact with host immune-inflammatory factors and other environmental factors, resulting in different outcomes with reference to mood disorders.

The positive association between MDD and LRI is consistent with earlier studies [68,92,125,128] and is most likely a two-way relationship. Immune-inflammatory changes associated with specific pathogens can trigger MDD in vulnerable individuals [129,130,131], while MDD may be associated both with reduced immunity to specific respiratory pathogens [132,133]. It is also possible that MDD may be linked to LRI through shared risk or vulnerability factors, such as substance abuse, socioeconomic disadvantage, comorbid medical conditions such as diabetes mellitus, or exposure to environmental pollutants [134,135,136]. Sometimes the association is more complex and can even span generations; for example, maternal depression has been associated with an increased risk of LRI in children [137].

Regarding the association between URI and BD, there is prior serological and clinical evidence that certain types of viruses causing URI, such as the influenza viruses and coronaviruses, are associated with mood disorders, but this association is not specific to MDD or BD [68,72]. There are several case reports of manic episodes being triggered by infection with these viruses [65,66,67,138]. A recent systematic review also concluded that viral infections could serve as a trigger for acute mania, but not for depression; however, the authors of this review relied on case reports and could not establish a specific relationship with URI [139]. A more recent study failed to find an association between bipolar disorder and the presence of antibodies to seasonal coronaviruses [140], which may be due to the milder nature of the infections caused by these viruses.

In contrast to the above findings, there is no specific evidence linking BD to an increased risk of or vulnerability to URI as a whole. Research conducted during the COVID-19 pandemic found that genetic vulnerability to BD was associated with susceptibility to this particular infection [141], and that patients with BD had worse outcomes when infected with SARS-CoV-2 [142]. Therefore, it is possible that such a relationship may exist for certain groups or types of viruses associated with URI, but this remains to be confirmed.

### 4.3. Relationship between the Current Results and Existing Hypotheses

The current study was not designed to specifically test any of the existing hypotheses linking infectious diseases and mood disorders. However, some of the results obtained can be interpreted as either supporting or contradicting existing hypotheses and as pointing toward improved or modified versions of these hypotheses.

First, the current results do not support a direct “protective” relationship between MDD and infectious diseases. If such a relationship existed, one would expect to find a negative correlation between population levels of MDD and at least some groups of infectious diseases [27]. No such relationship could be consistently demonstrated in the current data set, either in cross-sectional or longitudinal analyses. A modified version of this hypothesis postulates that depression can be triggered by certain infections. Once triggered, specific depressive symptoms, such as fatigue and social withdrawal, act to facilitate recovery and to minimize the risk of contagion or exposure to further infection [143]. This model cannot be either refuted or supported by the current data, as this would require an analysis of recovery and mortality rates from specific infections rather than incidence or prevalence.

Second, the possibility that at least some episodes of MDD can be triggered by infections, particularly respiratory infections, is supported by the current results in which changes in the incidence of URI and LRI were positively associated with changes in the incidence of MDD. Rantala et al. [33] suggested that MDD can be subtyped on evolutionary grounds based on specific triggering events such as infection, stress, trauma, bereavement, and other physical illnesses. They also highlighted the fact that short-term adaptive responses to these events can be maladaptive if prolonged, leading to syndromal MDD. They also suggested that this transition is more likely to happen in the setting of certain lifestyle factors, such as diet and insufficient physical activity, that can cause low-grade inflammatory activity. The PATHOS-D model is similar in many aspects to this model; though it does not specify any subtypes of depression, it is also based on the possibility of an ancient evolutionarily “adaptive” defense against infection turning “maladaptive” in response to social stressors or environmental circumstances [37]. The PATHOS-D and Rantala et al. models are broadly consistent with the current results. However, neither of them can account for the specific association between respiratory infections and mood disorders (MDD-LRI and BD-URI) identified in this study, and this relationship requires further exploration.

Third, these results provide mixed support for the “Old Friends” or hygiene hypothesis. If this model was applicable at the cross-national level, one would expect a negative correlation between intestinal infections and MDD, but this was not observed in the current study, though enteric infections were negatively correlated with BD. However, changes in enteric infections were negatively associated with changes in both MDD and BD. Therefore, it is possible that reduced exposure to enteric pathogens—which could serve as a proxy for exposure to “Old Friends”—is associated with an increased risk of mood disorders, and that this association may be—unexpectedly—stronger in BD than in MDD [144].

Fourth, the current study highlights an important limitation of all the above models, namely that they do not consider possible relationships between infectious diseases and bipolar disorder. Though some researchers have highlighted possible associations between both past and recent infections and BD [139,145], these findings have not been integrated into a broader model as has been attempted in the case of MDD. Bowins [53] suggested that milder forms of BD (hypomania) could represent a natural “override mechanism” that guards against the maladaptive aspects of depression. If this notion is placed alongside the PATHOS-D or Rantala et al. models, it is possible that hypomania (and possibly BD) would be more common in environments where the burden of infectious disease was relatively low, and the maladaptive aspects of depression would outweigh its benefits. A prior cross-national study found a positive correlation between national income and the prevalence of BD, which could be due, in part, to the lower incidence of infectious diseases in high-income countries [146]. This hypothesis, though tentative, is supported by the results of the current study, in which the incidence of BD was negatively correlated with all groups of infectious disease with the sole exception of URI. Longitudinal analyses also provide some support for this contention, as decreases in the incidence of enteric and tropical infections were associated with increases in the incidence of BD. Though the current results should be viewed as provisional, they suggest that it may be useful to expand evolutionary, infection-based models of MDD to incorporate BD or the entire category of mood disorders in general.

Finally, the above models notwithstanding, it is likely that any relationships between mood disorders and infectious diseases are bidirectional and complex. The cross-lagged panel analyses in our study did not support any direct, linear causal relationship between the two. It is likely that any association between infection and mood disorders may apply only to a subset of cases, may depend crucially on the specific properties of individual pathogens [29], and may be influenced by individual vulnerability factors as well as the social, cultural, and economic circumstances [147,148].

### 4.4. Possible Causal Mechanisms

The existence of significant associations between infectious diseases and mood disorders requires an evaluation of the possible mechanistic links between them. The most evident “point of contact” between these groups of disorders is the immune system, or, more broadly speaking, the neuro–immune–endocrine axis. Both MDD and BD are associated with altered levels of circulating cytokines and other inflammatory markers. There are several mechanisms through which these alterations can explain the links between infection and mood disorders. First, some of the immune changes seen in mood disorders may increase susceptibility to infection, as in the case of deficits in T and natural killer (NK) cell activity seen in MDD [97]. Second, certain vulnerability factors for depression, such as childhood adversity, can modify endocrine and immune activity, leading to elevations in “baseline” inflammatory activity. When such vulnerable individuals are exposed to infection, this could trigger an exaggerated or abnormal immune-inflammatory response, leading to an episode of MDD or BD [129,139,149,150]. Third, exposures to infection early in life, during childhood or adolescence, could “prime” the immune system of individuals with genetic variants affecting the expression of inflammatory mediators. This could lead to an exaggerated immune response and symptoms of MDD or BD in response to subsequent stressors, either infective or non-infective in nature [39,40,145,150,151]. These explanations are not mutually exclusive and are each influenced by other risk or protective factors for mood disorders, both genetic and environmental.

More recent research has drawn attention to the role of the microbiota–gut–brain axis in both MDD [152] and BD [153]. In the current study, enteric infections were negatively associated with the incidence of BD, and decreases in the incidence of enteric infections were associated with increases in both mood disorders, but more specifically with BD. These results are at variance with those of an earlier study, which found that intestinal infections were associated with an increase in subsequent mood disorders [76]. However, it is possible that variations in the incidence of enteric infections may be associated with variations in “non-pathogenic” components of the gut microbiota, which can influence symptomatology and treatment response in BD [154,155]. Alternately, exposure to certain pathogens could induce a state of “immune tolerance”, leading to a reduction in the immune-inflammatory dysregulation characteristic of mood disorders [156,157]. This would be a variation of the “hygiene hypothesis” [28,29,32]. A third possibility is that enteric infections could lead patients to receive treatment with antibiotics, which could alter the gut microbiome and trigger depressive or manic episodes [51,52,158,159]. The current results highlight the need to examine the links between enteric infections, gut microbiota, inflammation, and alterations in brain functioning in patients with mood disorders.

The current study also found positive associations between rates of respiratory infections and mood disorders, particularly between URI and BD and between LRI and MDD. These results point toward a third avenue that might link infectious and mood disorders, namely the “lung–brain axis”. This term refers to bidirectional interactions between the lungs and the brain, mediated through changes in the lung microbiome and the immune system as well as through direct neural links [160,161]. Alterations in the lung microbiome may influence levels of immune and inflammatory activity in the brain, and animal models have shown that infection with specific respiratory pathogens can induce brain inflammation and dysfunction of the blood–brain barrier [162,163]. Similar changes have been implicated in the pathogenesis of MDD and BD [164,165]. It is therefore possible that the results of our study, as well as earlier observations linking viral respiratory infections to mood disorders, could be explained in part by dysregulation of the lung–brain axis. This is particularly relevant in the context of the recent COVID-19 pandemic, which was accompanied by significant increases in the incidence of mood disorders in several parts of the world [166].

There is significant crosstalk between these axes at the molecular, cellular, and systemic levels [167]. This is depicted in Figure 2 above with specific reference to mood disorders. In summary, the relationship between infection and mood disorders may be influenced both by the specific properties of the invading micro-organism and by alterations in the functioning of the lung–brain, gut–brain and immune–endocrine–brain (psychoneuroendocrine) axes, leading to downstream effects such as brain inflammation, blood–brain barrier dysfunction, and increased sensitivity to stress [50,164,165,168]. It is unlikely that a simple one-to-one relationship exists between mood disorders and specific infections; rather, this relationship is mediated by a complex interaction between host, pathogen, and environmental variables, as well as through alterations in the aforementioned axes [169,170,171].

### 4.5. Strengths and Limitations

The strengths of the current study are its observational, hypothesis-free nature; its use of both cross-sectional and longitudinal methods; its use of the best available data from a large number of countries and territories; and its inclusion of both types of mood disorders. However, there are certain limitations inherent in the methodology adopted. First, as most of the analyses were of an associational nature, it was not possible to draw any definitive conclusions regarding causality. Second, though an attempt was made to assess causality using cross-lagged panel analyses, there are certain limitations inherent in this method [172]. Third, associations identified at the cross-national or regional level cannot be directly generalized to individuals [173]. Fourth, the relationship between infection and mood disorders is likely to be influenced by several confounding or interacting variables that could not be assessed in the current study, such as variations in the allele frequencies of immune-related genes, levels of social support and social capital, and antibiotic prescribing patterns [10,174,175]. Fifth, as infectious diseases were studied as groups, no link could be identified between a specific pathogen (such as the Epstein-Barr virus or *Toxoplasma gondii*) and mood disorders [55,176]. Likewise, other infectious disease categories that could be associated with mood disorders, such as periodontitis, could not be investigated using the current dataset [177]. Sixth, some researchers have suggested that links between infectious disease and mood disorders could be influenced by socioeconomic factors [32,44], but this could not be clearly established in the current study despite attempts to control for economic development and inequality. Seventh, it is possible that infectious diseases may non-specifically increase the risk for a wide range of mental disorders, and that the association with mood disorders is non-specific [56,76,178,179]. Finally, as the current study is preliminary and exploratory in nature, it was not yet possible to use the available data to estimate future trends in the association between infection and mood disorders [180].

## 5. Conclusions

The current study, based on analyses of data from the Global Burden of Disease studies, suggests that significant links exist between specific types of infectious diseases and mood disorders, particularly bipolar disorder. Though subject to the limitations mentioned above, these results provide some support for integrative, evolutionarily informed models, such as PATHOS-D, which link mood disorders to evolved defenses against infection. The current results also highlight the need for further investigation of the gut–brain and especially the lung–brain axes in the pathogenesis of mood disorders, as well as to extend models such as PATHOS-D to cover both major depression and bipolar disorder. Though preliminary in nature, this study points toward several promising avenues of future research. These include epidemiological and serological association studies of infectious diseases and mood disorders, cross-national studies examining the links between infection and mood disorders in individual subjects, and studies examining the interactions between infection and other risk or vulnerability factors, such as genetics, childhood adversity, environmental pollutants, and stress, in the pathogenesis of mood disorders. Such research could deepen our understanding of the causes of depression and mania and lead to innovative treatment approaches, such as those targeting gut or lung microbiomes or infection-triggered inflammatory pathways.

## Figures and Tables

**Figure 1 diseases-11-00116-f001:**
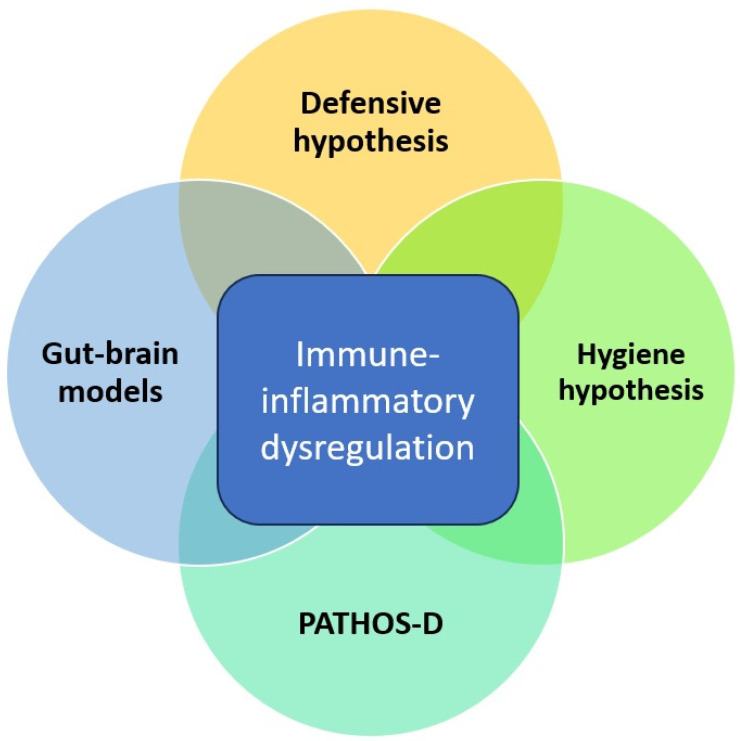
Overlap between evolutionary and mechanistic models linking infectious diseases and depression.

**Figure 2 diseases-11-00116-f002:**
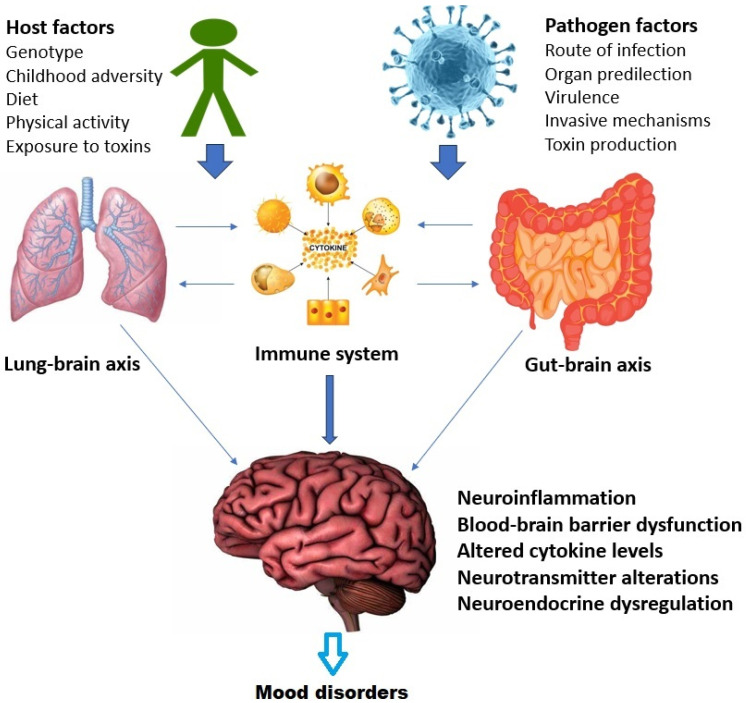
Interactions between the gut–brain, lung–brain, and immune systems in the relationship between infection and mood disorders.

**Table 1 diseases-11-00116-t001:** Descriptive statistics and changes over time for the incidence of mood disorders and infectious diseases.

Incidence	1990	2019	Change (%)	Significance
Major depressive disorder	7.48 (3.04)	7.10 (3.09)	−3.72 (10.56)	W = 15,547.5, *p* < 0.001
Bipolar disorder	0.10 (0.04)	0.10 (0.04)	0 (6.26)	W = 9039.5, *p* = 0.464
Upper respiratory infections	42.01 (9.28)	41.52 (8.51)	−0.80 (5.25)	W = 13,013.5, *p* = 0.002
Lower respiratory infections	1.59 (0.76)	1.14 (0.54)	−26.75 (12.13)	W = 20,903.0, *p* < 0.001
Enteric infections	16.19 (7.49)	19.08 (8.32)	12.49 (24.99)	W = 3686.5, *p* < 0.001
Tropical infectious diseases	184.03 (1127.66)	161.69 (812.85)	−20.70 (49.06)	W = 13,051.5, *p* < 0.001
Other infectious diseases	1.33 (0.89)	1.01 (0.28)	−21.77 (13.58)	W = 20,904.0, *p* < 0.001

Note: All incidence rates for 1990 and 2019 are given as median (inter-quartile range). Abbreviations: W, Wilcoxon’s signed-rank sum test statistic; *p,* significance level (*p*-value).

**Table 2 diseases-11-00116-t002:** Cross-sectional correlations between the incidence of mood disorders and infectious diseases, 1990–2019.

Year	Mood Disorder	URI	LRI	Enteric	Tropical	Other
1990	MDD	−0.02 (0.824)	0.08 (0.279)	−0.06 (0.430)	−0.02 (0.748)	−0.01 (0.971)
	BD	0.40 (<0.001) **	−0.31 (<0.001) **	−0.46 (<0.001) **	−0.29 (<0.001) **	−0.46 (<0.001) **
1995	MDD	−0.02 (0.731)	0.09 (0.207)	−0.02 (0.729)	−0.02 (0.815)	0.01 (0.957)
	BD	0.38 (<0.001) **	−0.28 (<0.001) **	−0.42 (<0.001) **	−0.29 (<0.001) **	−0.48 (<0.001) **
2000	MDD	−0.05 (0.523)	0.07 (0.321)	−0.02 (0.746)	−0.01 (0.919)	0.05 (0.461)
	BD	0.37 (<0.001) **	−0.29 (<0.001) **	−0.44 (<0.001) **	−0.32 (<0.001) **	−0.46 (<0.001) **
2005	MDD	−0.07 (0.313)	0.10 (0.146)	0.05 (0.475)	0.03 (0.675)	−0.01 (0.992)
	BD	0.36 (<0.001) **	−0.32 (<0.001) **	−0.43 (<0.001) **	−0.31 (<0.001) **	−0.51 (<0.001) **
2010	MDD	−0.06 (0.436)	0.11 (0.105)	0.06 (0.425)	0.03 (0.667)	0.02 (0.734)
	BD	0.37 (<0.001) **	−0.29 (<0.001) **	−0.44 (<0.001) **	−0.32 (<0.001) **	−0.50 (<0.001) **
2015	MDD	−0.07 (0.357)	0.12 (0.082)	0.07 (0.293)	0.04 (0.587)	0.05 (0.501)
	BD	0.38 (<0.001) **	−0.35 (<0.001) **	−0.44 (<0.001) **	−0.33 (<0.001) **	−0.51 (<0.001) **
2019	MDD	−0.05 (0.525)	0.11 (0.115)	0.07 (0.309)	0.02 (0.747)	0.09 (0.214)
	BD	0.41 (<0.001) **	−0.35 (<0.001) **	−0.46 (<0.001) **	−0.37 (<0.001) **	−0.43 (<0.001) **

Note: All correlations are provided as Spearman’s ρ (*p*-value). ** Significant at *p* < 0.05 after Bonferroni’s correction for multiple comparisons. Abbreviations: MDD, major depressive disorder; BD, bipolar disorder; URI, upper respiratory infections; LRI, lower respiratory infections.

**Table 3 diseases-11-00116-t003:** Partial correlations between the incidence of mood disorders and infectious diseases, 1990 and 2019, adjusted for possible confounding factors.

Year	Mood Disorder	URI	LRI	Enteric	Tropical	Other
1990	MDD	0.12 (0.209)	−0.04 (0.670)	−0.20 (0.036) *	−0.25 (0.009) *	−0.26 (0.006) *
	BD	0.00 (0.963)	0.03 (0.735)	−0.07 (0.481)	−0.21 (0.026) *	−0.42 (<0.001) **
2019	MDD	0.04 (0.650)	−0.09 (0.269)	−0.04 (0.607)	−0.14 (0.074)	−0.08 (0.307)
	BD	0.11 (0.158)	−0.18 (0.026) *	−0.23 (0.003) *	−0.29 (<0.001) **	−0.30 (<0.001) **

Note: All correlations are provided as Spearman’s partial ρ (*p*-value). All correlations are adjusted for the Human Development Index (HDI), Gini coefficient, percentage of population residing in urban areas, and distance from the equator. Full descriptions of these variables are given in Section 2.2. * Significant at *p* < 0.05, uncorrected. ** Significant at *p* < 0.05 after Bonferroni’s correction for multiple comparisons. Abbreviations: MDD, major depressive disorder; BD, bipolar disorder; URI, upper respiratory infections; LRI, lower respiratory infections.

**Table 4 diseases-11-00116-t004:** (a) Cross-lagged regression analyses of the relationship between mood disorders and infectious diseases, 1990–2019. (b) Cross-lagged regression analyses of the relationship between mood disorders and infectious diseases, 1990–2019, with outliers excluded.

(**a**)				
**Infectious Disease** **Category**	**Cross-Correlation (Infectious Disease 1990 × Mood** **Disorder 2019)**	**Cross-Correlation (Mood Disorder 1990 × Infectious** **Disease 2019)**	**Cross-Lagged Regression Coefficient**	**Significance Level**
URI				
× MDD	−0.08	−0.05	−0.029	0.681
× BD	−0.34	−0.32	0.018	0.798
LRI				
× MDD	0.12	0.03	0.097	0.168
× BD	−0.34	−0.34	0.002	0.977
Enteric				
× MDD	0.01	0.01	0.001	0.989
× BD	−0.43	−0.44	0.008	0.910
Tropical				
× MDD	0.22	0.14	0.075	0.286
× BD	−0.23	−0.27	0.038	0.589
Other				
× MDD	0.11	0.04	0.065	0.356
× BD	−0.43	−0.39	−0.037	0.599
(**b**)				
**Infectious Disease** **Category**	**Cross-Correlation (Infectious Disease 1990 × Mood** **Disorder 2019)**	**Cross-Correlation (Mood Disorder 1990 × Infectious** **Disease 2019)**	**Cross-Lagged Regression Coefficient**	**Significance Level**
URI				
× MDD	−0.26	−0.22	−0.040	0.644
× BD	0.14	0.14	−0.001	0.991
LRI				
× MDD	0.20	0.07	0.133	0.123
× BD	−0.20	−0.26	0.056	0.517
Enteric				
× MDD	0.16	0.14	0.022	0.799
× BD	−0.31	−0.29	−0.017	0.844
Tropical				
× MDD	0.27	0.22	0.048	0.579
× BD	−0.19	−0.24	0.056	0.517
Other				
× MDD	0.20	0.16	0.033	0.703
× BD	−0.38	−0.34	−0.041	0.636

Note: The analysis in Table 4b includes 136 countries. Overall, 68 countries were excluded due to zero or near-zero incidence estimates for certain infectious disease categories. Abbreviations: MDD, major depressive disorder; BD, bipolar disorder; URI, upper respiratory infections; LRI, lower respiratory infections.

**Table 5 diseases-11-00116-t005:** Correlations between percentage changes in the incidence of infectious diseases and mood disorders, 1990–2019.

Diagnostic Category	URI	LRI	Enteric	Tropical	Other
MDD	0.17 (0.013) *	0.29 (<0.001) **	−0.17 (0.013) *	−0.21 (0.002) *	0.13 (0.066)
MDD (adjusted)	0.06 (0.423)	0.27 (<0.001) **	−0.11 (0.176)	−0.16 (0.045) *	0.13 (0.108)
BD	0.92 (<0.001) **	0.05 (0.451)	−0.82 (<0.001) **	−0.29 (<0.001) **	−0.12 (0.084)
BD (adjusted)	0.88 (<0.001) **	0.00 (0.962)	−0.78 (<0.001) **	−0.23 (0.003) *	−0.08 (0.289)

Abbreviations: MDD, major depressive disorder; BD, bipolar disorder; URI, upper respiratory infections; LRI, lower respiratory infections. Note: All correlations are provided as Spearman’s ρ (*p*-value). Adjusted correlations are partial correlations adjusted for percentage changes in Gini coefficient and percentage of population residing in urban areas. * Significant at *p* < 0.05, uncorrected. ** Significant at *p* < 0.05 after Bonferroni’s correction for multiple comparisons.

**Table 6 diseases-11-00116-t006:** (a) Categorical associations between changes in the incidence of infectious diseases and changes in the incidence of major depression. (b) Categorical associations between changes in the incidence of infectious diseases and changes in the incidence of bipolar disorder.

(**a**)				
**Change in the Incidence of Infectious Disease**	**Change in the Incidence of Major Depression**		**χ^2^**	**Significance Level**
	**Decreased**	**Increased**		
URI				
Decreased	91 (65.5%)	29 (44.6%)	7.95	0.005 *
Increased	48 (34.5%)	36 (55.4%)		
LRI				
Decreased	138 (99.3%)	64 (98.5%)	0.31	0.537 ^†^
Increased	1 (0.7%)	1 (1.5%)		
Enteric				
Decreased	33 (23.7%)	22 (33.8%)	2.3	0.130
Increased	106 (76.3%)	43 (66.2%)		
Tropical				
Decreased	103 (74.1%)	54 (83.1%)	2.01	0.156
Increased	36 (25.9%)	11 (16.9%)		
Other				
Decreased	138 (99.3%)	63 (96.9%)	1.70	0.239 ^†^
Increased	1 (0.7%)	2 (3.1%)		
(**b**)				
**Change in the Incidence of Infectious Disease**	**Change in the Incidence of Bipolar Disorder**		**χ^2^**	**Significance Level**
	**Decreased**	**Increased**		
URI				
Decreased	102 (94.4%)	18 (18.8%)	120.22	<0.001 *
Increased	6 (5.6%)	78 (81.3%)		
LRI				
Decreased	107 (99.1%)	95 (99.0%)	0.01	0.999 ^†^
Increased	1 (0.9%)	1 (1.0%)		
Enteric				
Decreased	5 (4.6%)	50 (52.1%)	58.12	<0.001 *
Increased	103 (95.4%)	46 (47.9%)		
Tropical				
Decreased	75 (69.4%)	82 (85.4%)	7.31	0.007 *
Increased	33 (30.6%)	14 (14.6%)		
Other				
Decreased	106 (98.1%)	95 (99.0%)	0.23	0.999 ^†^
Increased	2 (1.9%)	1 (1.0%)		

Abbreviations: MDD, major depressive disorder; BD, bipolar disorder; URI, upper respiratory infections; LRI, lower respiratory infections. * Significant at *p* < 0.05, ^†^ Fisher’s exact test.

**Table 7 diseases-11-00116-t007:** Repeated measures analyses of changes in the incidence of mood disorders, 1990–2019.

Disorder	Effect	Test Statistic (*F*)	Significance Level
MDD	Time	26.51	<0.001 *
	Time × URI	1.07	0.349
	Time × LRI	0.02	0.982
	Time × Enteric	1.23	0.296
	Time × Tropical	0.78	0.468
	Time × Other	1.07	0.348
BD	Time	7.80	<0.001 *
	Time × URI	113.6	<0.001 *
	Time × LRI	4.55	<0.001 *
	Time × Enteric	49.00	<0.001 *
	Time × Tropical	1.11	0.356
	Time × Other	5.69	<0.001 *

Abbreviations: MDD, major depressive disorder; BD, bipolar disorder; URI, upper respiratory infections; LRI, lower respiratory infections. * Significant at *p* < 0.05 after Greenhouse-Gessler correction for sphericity.

**Table 8 diseases-11-00116-t008:** Non-linear curve fitting of the associations between infectious diseases and mood disorders, 1990 and 2019.

Disorder	Model with the Best Fit	1990		2019	
		R^2^	*p*	R^2^	*p*
MDD					
× URI	Linear	0.003	0.397	0.006	0.264
× LRI	Cubic	0.008	0.221	0.017	0.075
× Enteric	Linear	0.002	0.533	0.002	0.600
× Tropical	Linear	0.026	0.021 *	0.040	0.004 *
× Other	Linear	<0.001	0.868	<0.001	0.732
BD					
× URI	Quadratic	0.12	<0.001 *	0.12	<0.001 *
× LRI	Logarithmic	0.11	<0.001 *	0.13	<0.001 *
× Enteric	Quadratic	0.19	<0.001 *	0.23	<0.001 *
× Tropical	Logarithmic	0.07	<0.001 *	0.11	<0.001 *
× Other	Linear	0.26	<0.001 *	0.25	<0.001 *

Abbreviations: MDD, major depressive disorder; BD, bipolar disorder; URI, upper respiratory infections; LRI, lower respiratory infections; R^2^, percentage of variance explained; *p*, statistical significance level. * Significant at *p* < 0.05.

## Data Availability

The complete dataset used for the purposes of this study will be made available by the author on request.

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
