# Peer review of "Examining the Relationships between the Incidence of Infectious Diseases and Mood Disorders: An Analysis of Data from the Global Burden of Disease Studies, 1990–2019"

_diseases, 2023, doi:10.3390/diseases11030116_

Round 1

Reviewer 1 Report

Review of manuscript 2530015 by Ravi Philip Rajkumar

This study is an analysis of data from the Global Burden of Disease Studies, 1990-2019, examining the association between infectious diseases and mood disorders, specifically Major Depressive Disorder (MDD) and Bipolar Disorder (BD).

The introduction lays out the background literature in infectious disease and mood disorders and presents a comprehensive overview of theoretical frameworks linking the two. The literature review is balanced and serves as a useful introduction to the topic.

The materials and Methods are clearly laid out, and the analysis methodology is sound. Inclusion of potential confounding factors, and the utilization of three different approaches for longitudinal analyses are strong aspects of the study.

The results are presented clearly and comprehensively, covering both cross-sectional and longitudinal associations.

The discussion of results is extensive and thoughtful. Overall, this is an excellent contribution to literature.

The manuscript was well-written. I did notice several typographical or grammatical errors in the abstract and introduction. A close review of those sections should ameliorate any further concerns.

Author Response

This study is an analysis of data from the Global Burden of Disease Studies, 1990-2019, examining the association between infectious diseases and mood disorders, specifically Major Depressive Disorder (MDD) and Bipolar Disorder (BD).

The introduction lays out the background literature in infectious disease and mood disorders and presents a comprehensive overview of theoretical frameworks linking the two. The literature review is balanced and serves as a useful introduction to the topic.

The materials and Methods are clearly laid out, and the analysis methodology is sound. Inclusion of potential confounding factors, and the utilization of three different approaches for longitudinal analyses are strong aspects of the study.

The results are presented clearly and comprehensively, covering both cross-sectional and longitudinal associations.

The discussion of results is extensive and thoughtful. Overall, this is an excellent contribution to literature.

The manuscript was well-written. I did notice several typographical or grammatical errors in the abstract and introduction. A close review of those sections should ameliorate any further concerns.

Response: I thank the reviewer for their thoughtful and encouraging comments on my manuscript. I have proofread the original manuscript carefully and corrected these errors in spelling and grammar.

Reviewer 2 Report

It is not clear to me how the discussion of MDD and BD related to infectious diseases and why the authors were interested in the association between them. The introduction section spent a lot on the risk factors that potentially affect MDD and/or BD, and only very little background to introduce the main objective of this paper. And why these specific infectious diseases were considered only? Were the discussions limited by the data sources or something else?

The purpose of the subgroup analyses about the association between different infectious diseases was not clear enough, were these the secondary analysis? Otherwise, how do these contribute to the discussion of the association between MDD/BD and infectious diseases?

Author Response

I thank you for your thoughtful and insightful review of my paper. I have made revisions to the original manuscript as per your suggestions. These are described below:

1. It is not clear to me how the discussion of MDD and BD related to infectious diseases and why the authors were interested in the association between them. The introduction section spent a lot on the risk factors that potentially affect MDD and/or BD, and only very little background to introduce the main objective of this paper.

Response: I agree with the reviewer’s critique. The introduction has been rewritten extensively to discuss the existing evidence linking infectious diseases and mood disorders, covering both theoretical and epidemiological / clinical findings. Details of evolutionary, mechanistic and empirical perspectives, based on the latest published studies, have been included in the revised manuscript in Sections 1.1 to 1.3 (Lines 57-291). The rationale for the study has been clearly outlined in the new Section 1.4 (Lines 295-320). The section on other risk factors has been shortened to a brief section of the introductory paragraph.

2. And why these specific infectious diseases were considered only? Were the discussions limited by the data sources or something else?

Response: I apologize for omitting this important detail from the original manuscript. The rationale for selection of these specific disease groups was based on prior published data and has been described extensively in the revised manuscript in Section 1.3 (Lines 211-291), including citations of both past and recent research. Possible limitations of this approach have also been acknowledged in the Limitations section (lines 951-955 of the revised manuscript).

3. The purpose of the subgroup analyses about the association between different infectious diseases was not clear enough, were these the secondary analysis? Otherwise, how do these contribute to the discussion of the association between MDD/BD and infectious diseases?

Response: I agree with the reviewer that this section was not clear and was too long in the original manuscript. It has been clearly stated in the revised manuscript that this is a subgroup analysis and that it was carried out to address possible multicollinearity between infectious disease groups. Due to high multicollinearity between enteric and intestinal nematode infectious, and a lack of association between nematode infections and mood disorders in the literature, the later category was removed from the analyses in the revised paper. The remaining analyses have been described briefly in the text (Section 3.4.2) and the complete details have been moved to the Supplementary Material as they are not the primary objective of this study.

Reviewer 3 Report

Here are my comments:

1. Introduction:

1.1. The author needs to include a conceptual framework to support the rationale of this study.

2. Materials and Methods

2.1. Data sources: why choose 1990 and 2019 only? Is there a reason?

2.2. Power analysis of the study needs to include.

2.3. The author used regression / correctional analysis may not be able to handle data (e.g., HDI, Gini coefficient) with a time-series nature. Please elaborate on this issue or suggest employing other analysis methods to handle time-series data (e.g., GMM).

3. Discussion:

3.1. In the Strength and limitation section, the author wrote, "...it is possible that some of the associations between infection and mood disorder could be nonlinear in nature.." then, why the author did not use nonlinear analysis (you are using regression) to address this issue, please explain this.

Author Response

I thank the reviewer for their thoughtful and in-depth critique of my original manuscript. I have made corrections to the revised manuscript as per their suggestions and these have been described below.

Introduction:

1.1. The author needs to include a conceptual framework to support the rationale of this study.

Response: I agree completely with the reviewer. A new conceptual framework discussing evolutionary, mechanistic, and epidemiological / clinical evidence linking infectious diseases and mood disorders has been added (Sections 1.1 to 1.3, lines 57-291) with citations of all relevant theoretical papers and original research. The rationale for the study has been clearly described in the revised manuscript in Section 1.4 (lines 295-320). A new figure (Figure 1) has been added to illustrate the convergence of all evolutionary and mechanistic models in this field.

2. Materials and Methods

2.1. Data sources: why choose 1990 and 2019 only? Is there a reason?

Response: This period was selected because GBD data is available for 1990-2019, and because a longer time interval is preferrable for cross-lagged panel analyses as mentioned in references 120 and 172. On reviewing the GBD data, annual rates of change were small, and including this data might increase the risk of false-positive findings. Therefore, keeping in mind the reviewer’s valuable suggestion, data was collected at 5-year intervals (1990, 1995, 2000, 2005, 2010, 2015, 2019) and included in the cross-sectional and general linear model analyses for the revised manuscript. This is described in lines 376-385 of the revised manuscript. New analyses involving this data are presented in Table 2 and Table 8.

2.2. Power analysis of the study needs to include.

Response: I agree with the reviewer and apologize for omitting this. Power / sample size analysis has been included in the revised manuscript in lines 422-426.

2.3. The author used regression / correctional analysis may not be able to handle data (e.g., HDI, Gini coefficient) with a time-series nature. Please elaborate on this issue or suggest employing other analysis methods to handle time-series data (e.g., GMM).

Response: The current study is exploratory in nature and not predictive, as it is still not sure whether there is a significant causal or even correlational link between infection and mood disorders. Due to this, the general method of moments could not be used in the current study. This is mentioned in the Limitations of the study, lines 960-963 and reference 180. However, to address the reviewer’s valid concerns about the limitations of regression / correlation, general linear models with sphericity corrections were used to assess time-by-infectious disease interactions, using the 5-year interval data. The details of these analyses are mentioned in the Methods (Section 2.3, lines 408-411) and the results thereof are presented in Section 3.3.4 (lines 572-587) and the new Table 7 of the revised manuscript.

3. Discussion:

3.1. In the Strength and limitation section, the author wrote, "...it is possible that some of the associations between infection and mood disorder could be nonlinear in nature.." then, why the author did not use nonlinear analysis (you are using regression) to address this issue, please explain this.

Response: I agree with the reviewer’s valuable suggestion. Given that prior ecological association studies have found non-linear models to be useful (reference 121), non-linear curve fitting analyses were carried out and included in the revised manuscript. Statistical details of the non-linear analyses are mentioned in the Methods (Section 2.3, lines 412-416) and the results of non-linear analyses have been included in Section 3.5 (lines 636-648) and the new Table 8 in the revised manuscript.

Round 2

Reviewer 2 Report

I agree to the present draft for publication.

Reviewer 3 Report

The authors have changed the MS in accordance with their replies, in particular, added the key sections of the replies to my comments into the MS, well done!